# Novel Nanobiocatalyst Constituted by Lipase from *Burkholderia cepacia* Immobilized on Graphene Oxide Derived from Grape Seed Biochar

Lays C. de Almeida [1,2], Erikles L. O. Andrade [2], Jefferson C. B. Santos [2], Roberta M. Santos [3], Alini T. Fricks [4], Lisiane dos S. Freitas [3], Álvaro S. Lima [1,2], Matheus M. Pereira [5,*] and Cleide M. F. Soares [1,2,*]

1   Institute of Technology and Research, Avenida Murilo Dantas 300, Aracaju 49032-490, Brazil
2   Tiradentes University, Avenida Murilo Dantas 300, Aracaju 49032-490, Brazil
3   Department of Chemistry, Federal University of Sergipe, Av. Marechal Rondon, s/n, Jd. Rosa Elze, São Cristóvão 49100-000, Brazil
4   Department of Bromatological Analysis, Faculty of Pharmacy, Federal University of Bahia, Av. Barão de Jeremoabo s/n, Salvador 40170-115, Brazil
5   CICECO–Aveiro Institute of Materials, Chemistry Department, University of Aveiro, 3810-193 Aveiro, Portugal
*   Correspondence: matheus.pereira@ua.pt (M.M.P.); cleide18@yahoo.com.br (C.M.F.S.)

**Abstract:** The present research aims to study the process of immobilization of lipase from *Burkholderia cepacia* by physical adsorption on graphene oxide derived (GO) from grape seed biochar. Additionally, the modified Hummers method was used to obtain the graphene oxide. Moreover, Fourier transform infrared spectroscopy, Raman spectrum, X-ray diffraction, and point of zero charge were used for the characterization of the GO. The influences of pH, temperature, enzyme/support concentration on the catalytic activity were evaluated for the immobilized biocatalyst. The best immobilization was found ($543 \pm 5$ U/g of support) in the pH 4.0. Considering the biochemical properties, the optimal pH and temperature were 3.0 and 50 °C, respectively, for the immobilized biocatalyst. Reusability studies exhibited that the immobilized lipase well kept 60% of its original activity after 5 cycles of reuse. Overall, these results showed the high potential of graphene oxide obtained from biochar in immobilization lipase, especially the application of nanobiocatalysts on an industrial scale.

**Keywords:** lipase; biochar; graphene oxide

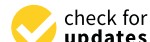



## 1. Introduction

The agro-waste requires the search for appropriate solutions for its destination in order to guarantee sustainable development and make this waste with greater added value. In this way, pyrolysis is one of the most used alternate ways for the use of agro-industrial waste, which consists of the thermal decomposition of biomass in the absence of oxygen, which generates bio-gas, bio-oil, and a solid carbon-rich residue (biochar) [1,2]. The biochar among the products generated is considered a multi-functional material in which it can be used for soil conditioner [3], adsorbent [4], support for enzyme immobilization [5], or as a precursor to activated carbon [6] and graphene oxide [7].

Graphene oxide (GO) can be produced from graphite by the Hummers method [8]. Graphene oxide showed great potential for application in different industrial sectors due to its properties such as high specific surface area, superior chemical resistance, good electrical and thermal conductivity, flexibility, and impermeability [9]. Among these applications, sensors, transparent conductive films, and enzymatic immobilization are included [10]. The use of immobilized biocatalysts is an alternative that enables their use in several industrial applications, as these catalysts can be easily separated from the product [10]. The recent development of nanostructured materials, such as graphene oxide (GO), that are materials with a greater surface area and different sizes and shapes, has enabled suitable support for

the immobilization of enzymes with different properties without compromising functionality, creating a novel nanobiocatalyst [11,12]. Different studies have been carried out to improve immobilization methods on organic and inorganic supports in order to reduce cost and improve thermal stability and performance of immobilized biocatalysts [12–14]. Accordingly, depending on the type of interaction between the enzyme and the support, it is possible to improve the biochemical, mechanical, and kinetic properties of the immobilized biocatalyst, and, for each type of support and enzyme, an appropriate immobilization method can be selected [15].

Therefore, the scientific community seeks alternatives of sustainable materials to support synthesis for enzyme immobilization utilizing nanobiocatalysts. In this sense, there are studies that used biomass residues to produce graphene oxide, as the study by Goswami et al. [16] that used rice straw biomass (RSB) to produce graphene oxide and applied in the adsorption of dyes. Thus, the present work aims to evaluate the potential of graphene oxide derived from grape seed biochar as a support for the immobilization of *Burkholderia cepacia* lipase (BCL) by physical adsorption (PA) using experimental data and enzyme protonation calculations. This study provided a theoretical basis with results for the future application immobilization of lipases on graphene oxide derived from biochar in the oil industry and others.

## 2. Materials and Methods

### 2.1. Materials and Reagents

The grape seed sample was kindly provided by Vitivinícola Quintas São Braz, located in the municipality of Petrolina, Pernambuco (Brazil). *Burkholderia cepacia* lipase (BCL) was purchased from Sigma Chemical Co. (St. Louis, MO, USA). Sulfuric acid P.A. (98%) was obtained from Vetec (São Paulo, Brazil); potassium permanganate P.A. (99%) was obtained from Vetec (São Paulo, Brazil); hydrogen peroxide P.A. (35%) and deionized (DI) water were obtained from Dinâmica (Rio de Janeiro, Brazil). The other chemicals were of analytical grade and used as received.

### 2.2. Biochar Production

The pyrolysis conditions for obtaining the grape seed biochar were performed on a laboratory scale as described by Santos et al. [17], with modifications. The conditions used were the following: temperature = 700 °C, heating rate = 30 °C min$^{-1}$, final time = 5 min and N$_2$ flow = 1 mL.min$^{-1}$.

### 2.3. Synthesis of Graphene Oxide

The synthesis of graphene oxide was carried out with some modifications as described by Goswami et al. [16]. The technique consists of contacting 5 g of the biochar in 100 mL of sulfuric acid (H$_2$SO$_4$) stored in a conical flask kept in an ice bath for a period of 1 h. After this period, 15 g of potassium permanganate was added slowly and left under stirring at room temperature for a period of 20 h. Then 400 mL of distilled water was added slowly and kept under stirring at 90 °C for a period of 1 h. Then 25 mL of hydrogen peroxide (H$_2$O$_2$) was added. After the system reached room temperature, the washing process was carried out with distilled water and then the material was taken to an oven at 60 °C.

### 2.4. Characterization of Graphene Oxide

To characterize the obtained graphene oxide, Fourier Transform Infrared Spectroscopy (FTIR) Analysis was performed. The samples of graphene oxide and biochar were analyzed on the FTIR Analysis equipment (BOMEM MB-100 FTIR Spectrophotometer). The spectra were obtained in the wavelength range from 400 to 4000 cm$^{-1}$. Raman spectra were recorded from 750 to 3500 cm$^{-1}$ on a Raman Renishaw 2000 Microprobe Confocal (Rhenishaw Instruments, England) using a 514.5 nm argon ion laser. For recording XRD patterns, a diffractometer with Cu K$\alpha$ radiation operating at 40 kV with 40 mA was used while 2$\theta$ values tested for ranged between 5 and 80°. Point of zero electric charge of the

graphene oxide was determined by a simple electrolyte addition method [18]. Briefly, 0.1 g of graphene oxide was immersed into 10 mL 0.05 M of different pH solutions and was shaken at a constant speed for 24 h, and the final pH of the solutions was measured.

### 2.5. Lipase Immobilization on Graphene Oxide by Physical Adsorption

The immobilization by physical adsorption (PA) of BCL on graphene oxide was accomplished by the methodology of Brito et al. [19] with little modifications. Adsorption experiments were conducted for 180 min at 25 °C with continuous stirring. The effect of pH and initial lipase concentration on the adsorption of lipase onto graphene oxide was investigated. pH values varied from 4.0, 6.0, 7.0,8.0 to 10.0 (buffer sodium acetate–pH 4.0 buffer sodium phosphate–pH 6.0, 7.0, 8.0 and buffer sodium carbonate–pH 9.0) at 0.1 M. Enzyme concentrations in $g_{enzyme}/g_{support}$ (0.15, 0.225, 0.3, 0.375 and 0.45) were studied in the immobilization process by physical adsorption to determine the best amount of lipase on the support.

### 2.6. Determination of Hydrolytic Activity

Hydrolytic activities were performed according to Soares et al. [20]. The entire assay was carried out under agitation, using 2 mL of used sodium phosphate buffer (0.1 M, pH 7.0), 100 mg of free or immobilized enzyme, and 5 mL of emulsion (olive oil with gum arabic solution (7% ($w/v$)), incubated for 10 min for immobilized or 5 min for free, at 37 °C. To interrupt the reaction, 2 mL of acetone:ethanol:water (1:1:1) was added. Phenolphthalein was applied as indicator for the titration process performed with potassium hydroxide solution (0.04 M). The evaluation of the hydrolytic activities performed for the lipase (free and immobilized) was used to determine the immobilization yield (%) according to Equation (1). Furthermore, it was defined that the amount of enzyme that released 1 μmol of free fatty acid per minute of reaction is equivalent to one unit (U) of enzymatic activity

$$\text{RI }(\%) = \frac{U_S}{U_0} \times 100 \tag{1}$$

where $U_S$ is the total enzyme activity recovered on the support and $U_0$ is the enzyme units offered for immobilization.

### 2.7. Computational Analysis

The Protein Data Bank (PDB) was used to obtain the crystal structure of BCL (PDB: 3 LIP). Water molecules and ligands were removed from the input PDB file, which was uploaded into the ProteinPrepare application (PlayMolecule web server–playmolecule.org) [21] to identify the titratable residue calculations. Computational analysis was performed at pHs 4, 7, and 10 to the pKa calculation. The output PDB files and protonation tables were downloaded and analyzed. The multigrid calculation of sequential focus automatically configured in the Adaptive Poisson–Boltzmann Solver (APBS) was used to determine the electrostatic properties.

### 2.8. Effect of pH and Temperature on Activity

The effect of pH on the activity of immobilized lipase was determined in buffer of values between pH 2.0 and 10. The buffers used were 0.1 M citric acid-sodium citrate (pH 2.0–5.0), 0.1 M potassium phosphate (pH 6.0–8.0), and 0.1 M bicarbonate-carbonate (pH 9.0–10). The optimal temperature for activity of immobilized lipase was assayed in the 25–80 °C range in the same 0.1 M potassium phosphate buffer (pH 3.0).

### 2.9. Thermal Stability and Reusability

The thermal stability of immobilized lipase was determined by incubating the biocatalyst in sodium phosphate buffer solutions (0.1 M, pH 3.0) for 3 h (with sampling each 1 h) at 50 °C. The reusability of the immobilized systems was assayed by running hydrolysis reactions in consecutive batches using the same biocatalyst. The time of each hydrolysis

reactions was 10 min at a temperature of 50 °C and pH of 3.0. After each reaction, the biocatalyst was rinsed once with hexane and reused for the next cycle of hydrolysis and the activity of the biocatalyst after the first cycle was considered 100%.

## 3. Results and Discussion

### 3.1. Characterization of Graphene Oxide

The FTIR spectrum of GO from grape seed biochar (Figure 1) that was observed was similar to the FTIR spectrum of the GO from commercial graphite. The strong peak around 3400 cm$^{-1}$ can be attributed to the O–H stretching vibrations of the C–OH groups and water [22], band in the 2970 cm$^{-1}$ region that are associated with the symmetric elongation of CH$_2$ [23], the C=C=O stretching vibrations at 2359 cm$^{-1}$ [7], the vibration band of epoxy group (1300 cm$^{-1}$) [7,23], as well as the strong band located at 1151 cm$^{-1}$ (C–OH stretching vibrations) [24].

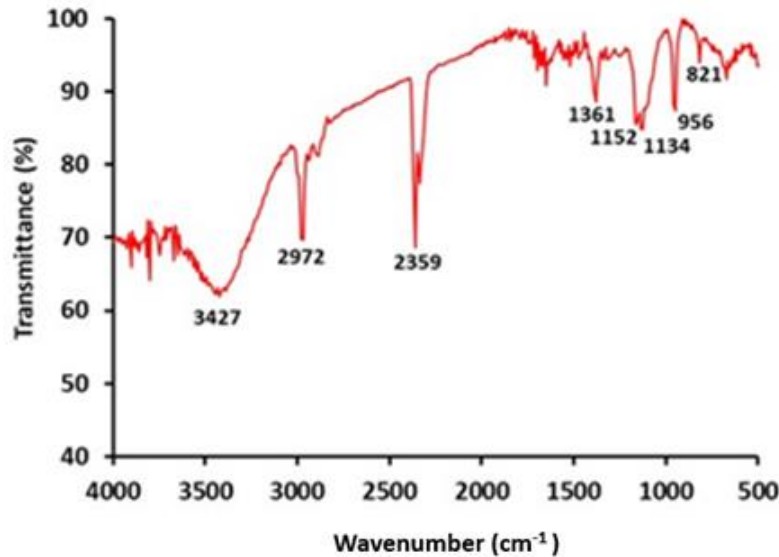

**Figure 1.** Fourier transform infrared spectroscopy (FTIR) spectrum of graphene oxide.

The Raman spectrum and X-ray diffraction (XRD) of graphene oxide synthesized from grape seed biochar are displayed in Supplementary Materials (Figures S1 and S2). The Raman spectrum displays two peaks at 1350 and 1586 cm$^{-1}$ denominated as the band D and band G, respectively. These bands are related to the structural defects and the sp$^2$ graphitized structure, respectively. These Raman results are observed in typical graphene oxide materials, which is in line with past studies [7,24]. The X-ray diffraction (XRD) of graphene oxide synthesized from grape seed biochar is shown the peak at 2θ = 11.6°, and the interlayer distance of 0.79 nm is in accordance with the results of previous studies [24].

The point of zero charge (pHzpc) was determined to investigate the surface charge of the graphene oxide. The graphene oxide derived from the biochar of seed grape showed acidic pHzpc, with a value of 2 (Figure 2). The determination of the pHzpc is important because the pH affects the adsorption process. In solutions with a pH below the point of zero charge, the surface of the graphene oxide is positively charged; consequently, the opposite behavior on the solution pH is greater than the pHzpc, and the surface of the graphene oxide is negatively charged [19].

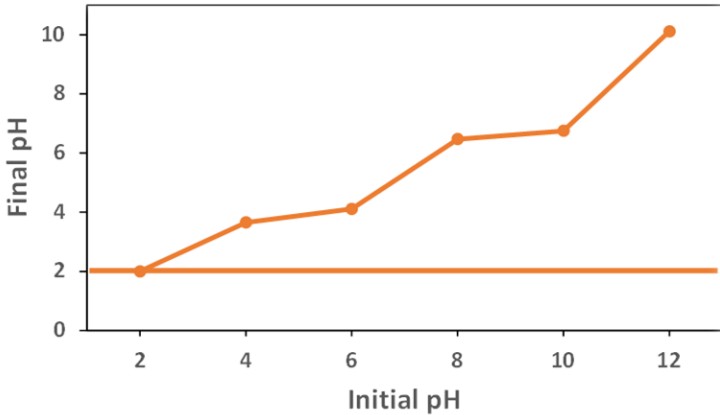

**Figure 2.** Point of zero charge for graphene oxide.

### 3.2. Effect of pH on Immobilization Process

The influence of the pH on the physical adsorption immobilization process of the BCL on the GO is shown in Figure 3. The pH of the solution changes the surface charge of the support and the electrostatic charge on the surface of the BCL, showing a significant parameter that affects the immobilization process. Higher activity in the derivative ($543 \pm 52$ U/g of support), displayed in Supplementary Materials (Table S1), was observed at a pH 4.0, showing a decreasing profile with increasing the pH. The point of zero charge, the pHzpc of graphene oxide, was found to be 2.0, and the surface is negatively charged when the solution pH is greater than the pHzpc. The isoelectric point of this lipase was 6.22 [25]. Consequently, the charges of the BCL residues are expected to be primarily positive at a a pH lower than 6.22 and negative at a higher pH. Consequently, the charges of the BCL residues are found to be mainly positive at an acidic pH and the contrary at an alkaline pH.

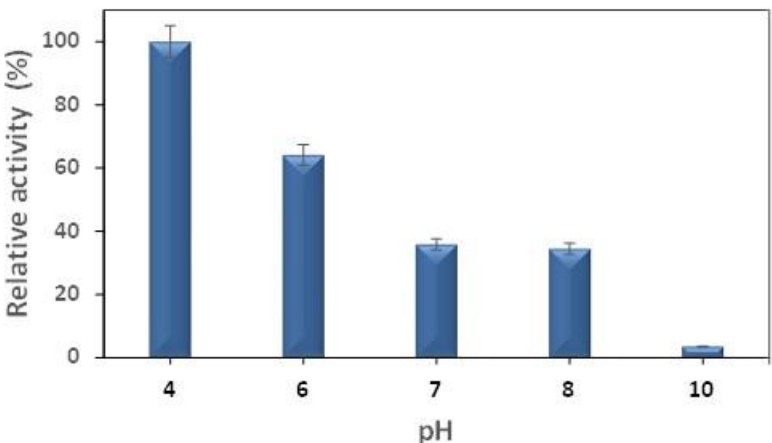

**Figure 3.** Effect of pH variation on BCL immobilization process.

According to experimental results obtained, the effect of the pH on the BCL immobilization showed that lower pHs (acidic environment) increase the enzyme adsorption capacity. On the other hand, at higher pHs (>7) the enzyme adsorption decreases, reaching the lowest value at pH 10. Thus, to obtain information about the BCL charge at various pHs, the enzyme protonation states were calculated. Figure 4 shows the charge distribution along the structure of the BCL, composed of a sequence of 320 amino acids, at pH 4, 7, and 10. At pH 4, below the isoelectric point of the enzyme, the identification of a quantity of 42 positively charged amino acids (Table S2), corresponds to the blue areas in Figure 4. With the increase in the pH, it is possible to identify the variation in the amounts of charges distributed in the structure, and at pH 7 (slightly above the isoelectric point) there is a decay of the total value of the positive charges to 32, causing a decrease in the blue surface

of Figure 4. For the pH 10 range, it is possible to notice that the enzyme surface presents mainly negative charges (red areas), totaling 29 positive charges only.

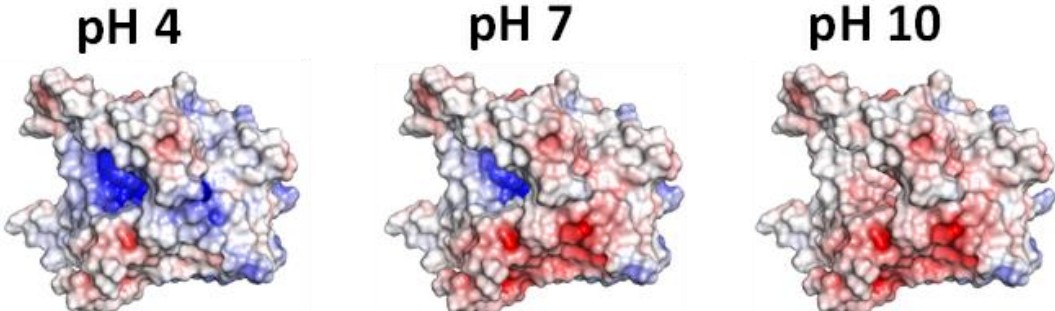

**Figure 4.** Electrostatic charge at the BCL surface calculated for pHs 4, 7, and 10. Red–white–blue scale refers to minimum ($-5$ kT/e, red) and maximum (5 kT/e, blue) surface potential.

Thus, according to the results obtained, it became possible to prove the influence of protonation in the immobilization process, showing that the capacity of attraction and repulsion of the support charges with the BCL surface charges influences the result of the immobilized biocatalyst. Similar results have been reported in the immobilization of the TLL adsorption on Amino–SiO2 [26] and lipase from porcine pancreas type II onto the activated carbons [27].

### 3.3. Effect of Enzyme/Support Ratio on Immobilization Process

To determine the best enzyme/support concentration, the results obtained were calculated according to the relative activity. Figure 5 shows that the maximum relative hydrolytic activity of the BCL immobilized by the physical adsorption was obtained with 0.375 $g_{enzyme}/g_{support}$ with about a 36% immobilization yield and 457 U/g hydrolytic activity; the others are displayed in the Supplementary Materials (Table S3). The profile obtained may be due to the saturation of the support with the lipase, showing a decrease in activity above 0.375 $g_{enzyme}/g_{support}$. This decrease in relative activity can be attributed to the formation of a multilayer protein structure blocking or inhibiting access to the enzyme's active sites [28]. Therefore, the concentration of 0.375 g was selected for the following stages of this work. Enzymatic adsorption on the graphene oxide occurs through weak bonds, such as van der Waals forces, ionic interactions, or hydrophobic interactions [29].

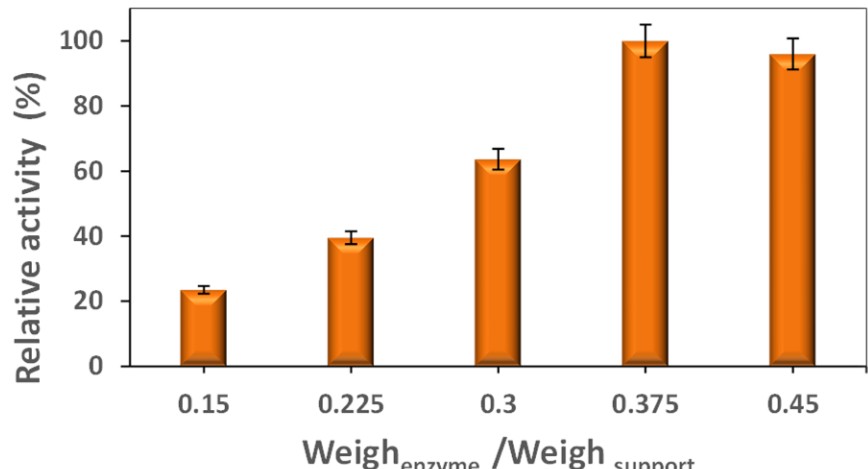

**Figure 5.** Relative activity (%) as a function of enzyme/support ratio during immobilization of the BCL on graphene oxide.

Similar behavior was observed in the literature [28–31] regarding the enzyme concentration variation on the immobilization study in different tips of support. The same

profile was found for Zang et al. [31] in the study with graphene oxide for the immobilization for ROL. The highest immobilization yield was obtained at low ROL concentrations (100–500 µg mL$^{-1}$), with an increasing soluble ROL concentration (7.5–15.0 mg mL$^{-1}$). A significant drop in ROL yield immobilization was observed, and the values were the same for the two supports studied. Activity recovery drastically decreased with increasing the ROL concentration, which could be a consequence of protein–protein interactions occurring at a high enzyme load.

Table 1 presents the enzymatic load values (g/g of support) and activity recovery (%) of the different enzymes and supports. This work using graphene oxide produced using grape seed biochar, which is a material derived from agroindustrial waste, gives this waste an added value and an environmentally friendly destination. The production of biochar is carried out by pyrolysis, which also generates bio-gas and bio-oil of industrial interest. In this study, the immobilization of LBC by physical adsorption on graphene oxide produced from grape seed biochar showed a recovered activity of 100% % for this work, which shows to be promising support when compared to other studies that use graphene oxide [32] or magnetic graphene oxide [33,34] that show a recovered activity of less than 100%. Other types of supports used for the immobilization of lipases showed recovery activity less than 100%.

**Table 1.** Enzyme loading values (g/g of support) and activity recovery (%) of different enzymes and supports.

| Enzyme | Support | Activity Recovery (%) | Enzyme Loading (g/g support) | Ref. |
|---|---|---|---|---|
| Lipase from *Burkholderia cepacia* | Graphene oxide derived from grape seed biochar | 100 | 0.375 | This Work |
| Lipase from *Rhizopus oryzae* | Graphene oxide | 25 | 0.02 | [32] |
| Lipase from *Thermomyces lanuginosa* | Magnetic Fe$_3$O$_4$ nano-particles | 70 | 0.250 | [33] |
| Lipase from *Candida rugosa* | Graphene oxide encapsulated Fe$_3$O$_4$ | 64 | 0.02 | [34] |
| Lipase from *Thermomyces lanuginosa* | Graphene oxide functionalized with lysine | 150 | 0.115 | [35] |
| Lipase from *Burkholderia cepacia* | Resin NKA | 96 | 0.110 | [36] |
| Lipase from *Candida rugosa* | Magnetic microspheres with hydrophilicity | 64 | 0.100 | [37] |
| Lipase from *Burkholderia cepacia* | PST microspheres | 50 | 0.252 | [38] |

Zhou et al. [35] used GOs functionalized with amino acids for the immobilization of lipase from *Thermomyces lanuginosa* adsorption, and the results showed an improvement in the enzymatic activity, obtaining a recovered activity of 150%. This demonstrates that the functionalization of GOs can increase the activity recovered from the present work that uses graphene oxide from grape seed biochar. Thus, future work will be carried out with the aim of increasing the activity recovered from the present study using other enzyme immobilization protocols.

### 3.4. Effects pH and Temperature of the Immobilized BCL on Graphene Oxide

Figure 6 shows the effect of the pH on the lipase activity of the immobilized BCL. The optimum pH for the free lipase from *Burkholderia cepacia* is 7.0 [27]. It can be noted that for the immobilized BCL, the ideal pH was changed to a more acidic pH of 3.0 (462 U/g). Other pHs are provided in the Supplementary Materials (Table S4). Changes in the ideal pH for the immobilized enzymes have been reported for the different sources of lipase and types of transporters [29]. A study by Carvalho et al. [27] on *Burkholderia cepacia* lipase immobilized on xerogel silica showed a change in the pH optimum to 3.0, whereas the pH optimum of the free lipase was 7.0. This alteration was due to the properties of the support and the method of immobilization.

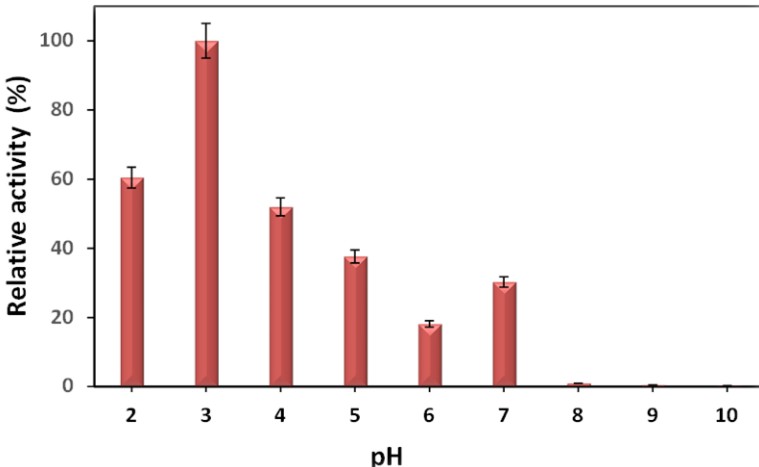

**Figure 6.** pH effect on the activity the immobilized BCL by physical adsorption (PA) onto graphene oxide.

In general, lipases immobilized on polycationic supports tend to change the optimal pH to more acidic values [39]. In the study by Pereira et al. [39], a change from the optimal pH of 7.0 of the free lipase of Candida rugosa to the optimal pH of 6.0 for the lipase immobilized by PA in chitosan was observed. This was due to imbalances in the microenvironment of the immobilized enzyme caused by the electrostatic interactions between the enzyme and the support. According to Abdulla and Ravindra [40], the surface of the enzymes contains numerous acidic and basic groups, and the charge of these groups changes depending on the pH of the medium, resulting in enzymatic activity, structural stability, and solubility may change.

The reaction speed depends on the temperature, which is a very important parameter. This is because, with increasing temperature, more kinetic energy is observed in the reactant molecules, increasing the number of productive collisions per unit of time. However, the enzyme-catalyzed reactions require that the tertiary and secondary structures of the enzyme remain intact to maintain the same catalytic activity. It is important to evaluate the effect of temperature on the microenvironment of the immobilized biocatalysts, as high temperature can lead to excess energy absorption, leading to disruption of this structure and consequent denaturation of the enzyme [41].

The effect of temperature on the catalytic activity of the immobilized biocatalyst can be seen in Figure 7. The free lipase of *Burkholderia cepacia* shows an optimum temperature to 50 °C [27]. The maximum activity of the immobilized biocatalysts occurred at 50 °C (661 U/g); the other temperature values are presented in the Supplementary Materials (Table S5). This was probably due to the process of immobilization of the lipase to the support, which reduces mobility by restricting contact with the substrate, resulting in a reduction in activity. In studies using inorganic supports for the BCL immobilization supported on mesoporous silica, maximum activity was observed at 50 °C [27]. Likewise, it was also seen in studies involving organic supports developed by Cabrera-Padilla et al. [42], who used a natural biopolymer poly(3-hydroxybutyrate-co-hydroxyvalerate) (PHBV) for the immobilization of Candida rugosa lipase, presenting an optimal immobilized temperature range of 37–45 °C. The immobilized bicatalizers showed a high hydrolytic activity at higher temperatures compared to the free enzymes, which can be suggested due to the more rigid conformation of the immobilized enzymes because of the electrostatic interactions and hydrogen bonds between the enzyme and the support [42].

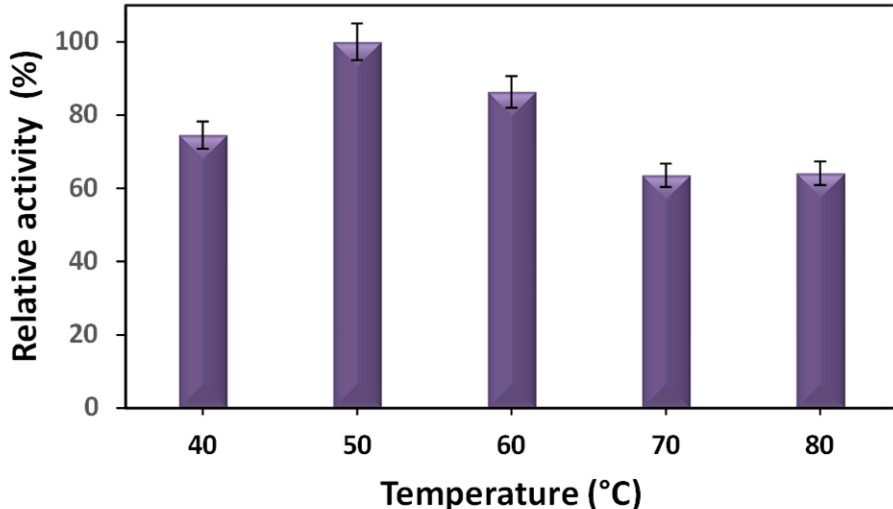

**Figure 7.** Temperature effect on the immobilized BCL by physical adsorption (PA) onto graphene oxide.

### 3.5. Thermal Stability and Operational Stability of the Immobilized BCL on Graphene Oxide

The thermal stability of the immobilized biocatalyst was investigated by incubating them at 50 °C in different time durations (30 to 180 min), and the evaluation of their residual activities is shown in Figure 8. The residual activity of the immobilized biocatalyst, which was incubated at 50 °C for 0 min, was considered as the control with 100% activity; the other values are displayed in the Supplementary Materials (Table S6). As shown in Figure 7, the residual activity of the immobilized biocatalyst was decreased by increasing the incubation time, and it maintained its residual activity above 50% after 120 min of incubation time.

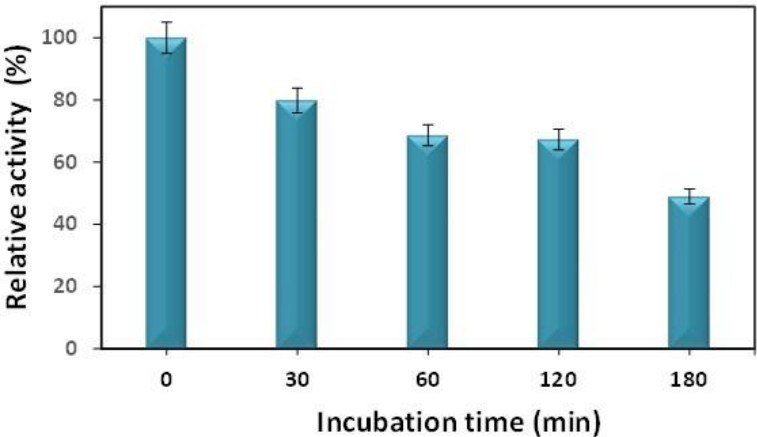

**Figure 8.** Thermal stability of the immobilized biocatalyst incubated at 50 °C.

According to the literature, this profile is reported in other works that used graphene oxide as support for enzyme immobilization. For example, the work carried out by Nematian et al. [30] used graphene oxide with different activation processes for the ROL immobilization and observed the same behavior with a reduction in residual activity with increasing the incubation time.

Operational stability is one of the main parameters evaluated for the application of the biocatalyst on an industrial scale, as the reuse of the immobilized enzyme reduces the cost of the process. The operational stability of LBC immobilized on graphene oxide was tested in an olive oil emulsion hydrolysis reaction for 10 min at 50 °C, as seen in Figure 9. LBC immobilized on graphene oxide by physical adsorption can be reused 5 times maintaining above 50% of its initial activity; the other values are shown in Supplementary Materials (Table S7).

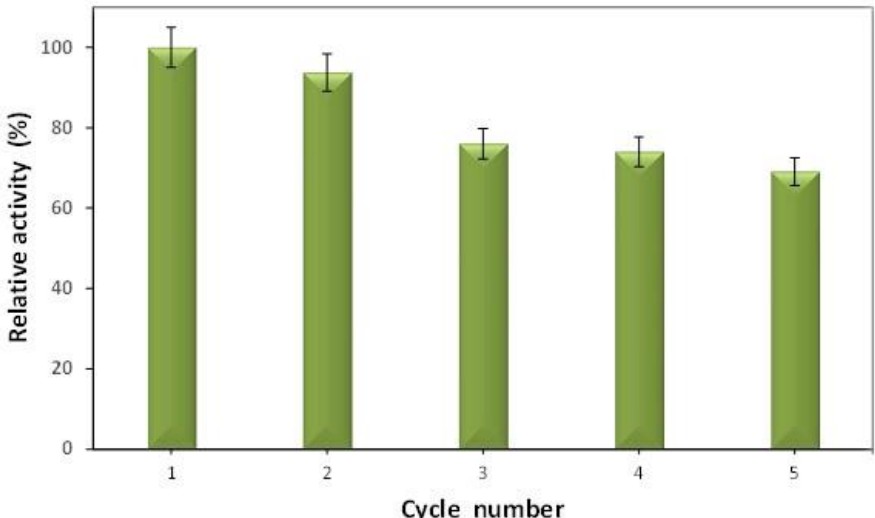

**Figure 9.** Operational stability of the immobilized biocatalyst.

This fact may be related to the hydrophobicity of graphene oxide. This is because lipases are characterized by a lid at the catalytic site that changes conformation depending on the environment, an open conformation at the water/oil interface, and a closed conformation for aqueous media. Knežević et al. [43] reported that the lipases recognize the hydrophobic surfaces similar to the natural substrates and are activated during immobilization at the interface, creating an open active site accessible to the substrates. This greatly enhances the activity of most lipases when adsorbed onto hydrophobic supports.

In addition, the immobilization of lipases by physical adsorption is simple and inexpensive, and it is often possible to regenerate the immobilized biocatalysts because the interactions between the enzyme and the support are mediated by direct interactions (mainly van der Waals, hydrogen bonds, and interactions hydrophobic). In recent research, the lipase was immobilized on the magnetic amino-functionalized graphene oxide nanocomposites. The reported results after 5 cycles were that 70% remained active for its temperature range, whose activity was about 40–60 °C, and for its recyclability [30].

## 4. Conclusions

In the present work, the use of grape seed biochar was described as an alternative to an ecological and low-cost material to produce graphene oxide to replace graphite in production using the modified Hummers method. Promising results were obtained for the process of immobilization of lipase from *Burkholderia cepacia* by physical adsorption on graphene oxide derived from grape seed biochar. The study of the immobilization parameters made it possible to evaluate the forces involved in the enzyme/support binding, thus establishing the best immobilization conditions for obtaining a biocatalyst with high hydrolytic activity. The graphene oxide obtained has potential to be employed in lipase immobilization, given that the immobilized enzymes showed good enzyme activity. Therefore, this promising biocatalyst shows new development opportunities for application in several industrial sectors.

**Supplementary Materials:** The following supporting information can be downloaded at: https://www.mdpi.com/article/10.3390/c9010012/s1, Figure S1:Raman spectrum of GO obtained from grape seed biochar. For Raman spectra were recorded from 750 to 3500 cm$^{-1}$ on a Raman Renishaw 2000 Microprobe Confocal (Rhenishaw Instruments, England) using a 514.5 nm argon ion laser.; Figure S2: X-ray diffraction (XRD) pattern of graphene oxide from grape seed biochar. For recording XRD patterns, a diffractometer with Cu Kα radiation operating at 40 kV with 40 mA was used while 2θ values tested for ranged between 5–80°.; Table S1: Effect of pH variation on BCL immobilization process; Table S2. Distribution of amino acids of the BCL and their respective charges for pH 4, 7, and 10; Table S3. Immobilization Yield (%), Hydrolytic activity (U/g) and Relative activity (%)

as a function of enzyme/support ratio during immobilization of the BCL on graphene; Table S4 pH effect on the activity the immobilized BCL by physical adsorption (PA) onto Graphene Oxide; Table S5. Temperature effect on the immobilized BCL by physical adsorption (PA) onto graphene oxide; Table S6. Thermal stability of the immobilized biocatalyst incubated at 50 °C.; Table S7. Operational stability of the immobilized biocatalyst.

**Author Contributions:** Investigation: A.T.F. and L.d.S.F.; methodology: L.C.d.A., E.L.O.A. and J.C.B.S.; validation: R.M.S.; supervision: Á.S.L., M.M.P. and C.M.F.S. All authors have read and agreed to the published version of the manuscript.

**Funding:** This research was funded by Fundação de Apoio à Pesquisa e Inovação Tecnológica do Estado de Sergipe FAPITEC/SE (PROMOB–Edital CAPES/FAPITEC/SE n° 01/2013) and Coordenação de Aperfeiçoamento de Pessoal de Nível Superior (CAPES) (88887357049/2019-00) for research scholarships.

**Data Availability Statement:** Not applicable.

**Acknowledgments:** The authors acknowledge financial support from Conselho Nacional de Desenvolvimento Científico e Tecnológico (CNPq), Coordenação de Aperfeiçoamento de Pessoal de Nível Superior (CAPES), Fundação de Apoio à Pesquisa e Inovação Tecnológica do Estado de Sergipe (FAPITEC) and Universidade Tiradentes (UNIT) and CLQM (Center of Multi-users Chemistry Laboratories) from Federal University of Sergipe for the analysis support.

**Conflicts of Interest:** The authors declare that there are no conflict of interest.

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
