# Peer review of "Novel Nanobiocatalyst Constituted by Lipase from Burkholderia cepacia Immobilized on Graphene Oxide Derived from Grape Seed Biochar"

_carbon, 2022_

Round 1

Reviewer 1 Report

The article «Novel nanobiocatalyst constituted by lipase from Burkholderia cepacia immobilized on graphene oxide derived from grape seed biochar» is attracted to a relevant topic and has a high applied value.

The present article aims to study the process of immobilization of lipase from Burkholderia cepacia by physical adsorption on graphene oxide derived (GO) from grape seed biochar. FTIR, Raman spectrum, X-ray diffraction and point of zero charge were used for the characterization of GO. The influences of pH, temperature, enzyme/support concentration on the catalytic activity were evaluated for the immobilized biocatalyst. These results showed the high potential of graphene oxide obtained from biochar in immobilization lipase, especially nanobiocatalysts application on an industrial scale.

Major comments:

1. The performed study should be a connected with determined enzyme catalytic constant such as Km, Vmax and Kcat.

2. Authors should add a table (as in https://doi.org/10.1016/j.ijbiomac.2019.07.132, Table 4) or figure (as in https://doi.org/10.3390/polym14235110, Figure 6) to the article containing the results of comparison of the optimized enzyme loading and activity recovery with the results of other authors and/or own results obtained earlier.

3. The authors write that, the Raman spectrum and X-ray diffraction (XRD) of graphene oxide synthesized from grape seed biochar, displayed in Supplementary Materials (Figure S1- S2), but does not discuss the results in any way in the text of the article. In addition, a detailed description of the methods FTIR, Raman spectrum, and XRD should be added to the materials and methods section.

4. It is necessary to indicate whether the amino acids from the Table S2 are located on the surface of the enzyme molecule. Ideally, select those that are located on the surface, because they are the ones that interact with the carrier during immobilization.

Minor remarks:

1. In the title of the article Cepacia for an unknown reason with a capital letter.

2. Line 79 and Line 85: SANTOS and GOSWAMI for an unknown reason highlighted in capital letters.

3. Line 229: dot missing.

4. Lines 236-237: the designation genzyme/gsupport looks strange.

5. Figure 5: the designation Weightenzima/Weightsuporte looks strange. Probably the authors forgot to translate the caption to the x-axis into English.

6. Lines 270-271: Candida rugosa should be italicized.

7. Line 329: strange spelling of the author's name – knežević et al.

Author Response

DETAILED ANSWER TO THE REVIEWER’S COMMENTS

REVIEWER 1

The article «Novel nanobiocatalyst constituted by lipase from Burkholderia cepacia immobilized on graphene oxide derived from grape seed biochar» is attracted to a relevant topic and has a high applied value. The present article aims to study the process of immobilization of lipase from Burkholderia cepacia by physical adsorption on graphene oxide derived (GO) from grape seed biochar. FTIR, Raman spectrum, X-ray diffraction and point of zero charge were used for the characterization of GO. The influences of pH, temperature, enzyme/support concentration on the catalytic activity were evaluated for the immobilized biocatalyst. These results showed the high potential of graphene oxide obtained from biochar in immobilization lipase, especially nanobiocatalysts application on an industrial scale.

Major comments:

  • Reviewer 1: The performed study should be a connected with determined enzyme catalytic constant such as Km, Vmax and Kcat.

Answer: We agree with the reviewer that a study to determine enzyme catalytic constants such as Km, Vmax and Kcat is important for enzyme immobilization studies. However, it was not the objective of this study. In future studies, these parameters will be performed

  • Reviewer 1: Authors should add a table (as in https://doi.org/10.1016/j.ijbiomac.2019.07.132, Table 4) or figure (as in https://doi.org/10.3390/polym14235110, Figure 6) to the article containing the results of comparison of the optimized enzyme loading and activity recovery with the results of other authors and/or own results obtained earlier.

Answer: we added a table in article containing the results of comparison of the optimized enzyme loading and activity recovery

(3) Reviewer 1:   The authors write that, the Raman spectrum and X-ray diffraction (XRD) of graphene oxide synthesized from grape seed biochar, displayed in Supplementary Materials (Figure S1- S2), but does not discuss the results in any way in the text of the article. In addition, a detailed description of the methods FTIR, Raman spectrum, and XRD should be added to the materials and methods section.

Answer: we added   the results the Raman spectrum and X-ray diffraction (XRD) in the text of the article and  a detailed description of the methods FTIR, Raman spectrum, and XRD  to the materials and methods section.

(4) Reviewer 1:   It is necessary to indicate whether the amino acids from the Table S2 are located on the surface of the enzyme molecule. Ideally, select those that are located on the surface, because they are the ones that interact with the carrier during immobilization.

Answer: We agree with the reviewer and acknowledge the recommendations. The table S2 highlighted in gray the amino acids that are located on the surface of the enzyme molecule

Minor remarks:

(1) Reviewer 1:   In the title of the article Cepacia for an unknown reason with a capital letter.

Answer: The article Cepacia have been changed to lower case

(2) Reviewer 1:    Line 79 and Line 85: SANTOS and GOSWAMI for an unknown reason highlighted in capital letters.

Answer: The names have been changed to lower case

  • Reviewer 1:  Line 229: dot missing.

Answer: Dot added

(4) Reviewer 1:    Lines 236-237: the designation genzyme/gsupport looks strange.

Answer: The designation genzyme/gsupport has been corrected to genzyme/gsupport.

(5) Reviewer 1:    Figure 5: the designation Weightenzima/Weightsuporte looks strange. Probably the authors forgot to translate the caption to the x-axis into English.

Answer: The designation Weightenzima/Weightsuporte  has been corrected to Weightenzyme/Weightsupport

(6) Reviewer 1:    Lines 270-271: Candida rugosa should be italicized.

Answer:  Candida rugosa has been italicized

(7) Reviewer 1:    Line 329: strange spelling of the author's name – knežević et al.

Answer: The spelling of the author's name – knežević et al. it is written correctly.  https://doi.org/10.2298/APT0435151K

Reviewer 2 Report

The manuscript presents the immobilization of lipase on graphene oxides derived from grapeseed biochar and evaluated its hydrolysis activity using olive oil. Enzyme immobilization has been extensively studied and below are my comments and questions that I hope would be adequately addressed by the authors:

1. Is there any specific characteristic of biochar derived from grape seed that the authors are looking at as compared to biochar derived from any other material? There has been extensive works on the enzyme immobilisation. What is the novelty presented in this paper?

2. The manuscript mainly reports experimental trends at different conditions with little discussions about the resulting trends.

3. The results from the immobilized enzyme should also be compared with those of the free enzymes followed with detailed discussions to compare between the two to give more insights to the readers.

4. What is the advantages/highlights of this BCL immobilized on GO? Is there any particular advantage of using graphene oxide as support as compared to using activated carbon, for example? Proper comparisons should also be drawn with other types supports to highlight the novelty/significance of this work. 

5. For Fig 9, Is the decrease of activity after recycle due to the changes in the enzyme structure or due to the leaching of the enzyme from the support?

6. The authors need to pay more attention to the grammar and the vocabulary used in the manuscript. For instance, in Fig 5, the X-axis label is written in Portuguese, not English.

Author Response

REVIEWER 2

The manuscript presents the immobilization of lipase on graphene oxides derived from grapeseed biochar and evaluated its hydrolysis activity using olive oil. Enzyme immobilization has been extensively studied and below are my comments and questions that I hope would be adequately addressed by the authors:

(1) Reviewer 2:    Is there any specific characteristic of biochar derived from grape seed that the authors are looking at as compared to biochar derived from any other material? There has been extensive works on the enzyme immobilisation. What is the novelty presented in this paper?

Answer:  Agroindustrial residues among the types of biomass available for the production of chemical and energy products can be considered the most economically attractive. Fruit seeds are a type of waste that has been reported as suitable for use as a feedstock for the production of biofuels. The chemical composition of the biomass influences the characteristics of the material. Currently, a broad collection of new supports for enzyme immobilization are coming up, allowing the researchers to specifi cally choose a la carte different features depending on the enzyme and the given application (e.g., particle size, chemical functionality, length of spacer arm, porosity, the hydrophile–lipophile balance of the microenvironment surrounding the enzyme, and more). In this way, it is extremely important to evaluate the different types of biomass as precursors for the production of biochar and to evaluate them as a support for the immobilization of enzymes.

Therefore, this study aimed to determine the best conditions for the synthesis of graphene oxide derived from grape seed biochar. the grape seed was used as a carbon precursor for the production of supports, since this waste is generated in large quantities in Brazil, mainly from the processing of grapes for the production of wines and juices. Furthermore, this study performed a detailed analysis of the lipase immobilization by adsorption on activated carbon, including evaluation of adsorption equilibrium, effect of pH, temperature, thermal stability, Reusability and computational analysis.

(2) Reviewer 2:  The manuscript mainly reports experimental trends at different conditions with little discussions about the resulting trends.

Answer: We agree with the reviewer and acknowledge the recommendations.

(3) Reviewer 2:  The results from the immobilized enzyme should also be compared with those of the free enzymes followed with detailed discussions to compare between the two to give more insights to the readers.

Answer: We agree with the reviewer and acknowledge the recommendations.  Free enzyme results added in discussions and compare between immobilized enzyme results

(4) Reviewer 2:  What is the advantages/highlights of this BCL immobilized on GO? Is there any particular advantage of using graphene oxide as support as compared to using activated carbon, for example? Proper comparisons should also be drawn with other types supports to highlight the novelty/significance of this work. 

Answer: Graphene oxide (GO) bears various oxygen-containing functional groups these abundant surface functional groups on the GO can be the key chemical skeletons that are used as immobilizing sites for pure lipase, facilitating efficient lipase immobilization through chemical bonds or electrostatic interactions. And lipase with primary amine groups can be readily immobilized on the surface of the GO with carboxyl groups. Future work will be carried out using other immobilization protocols such as covalent bonding with different spacer arms.

(5) Reviewer 2:  For Fig 9, Is the decrease of activity after recycle due to the changes in the enzyme structure or due to the leaching of the enzyme from the support?

Answer: The decrease in activity after recycling is due to the leaching of the enzyme from the support, as the method used for enzyme immobilization was physical adsorption, which is a method that has low binding strength. Future studies will be carried out using another method of enzymatic immobilization such as covalent binding.

(6) Reviewer 2: The authors need to pay more attention to the grammar and the vocabulary used in the manuscript. For instance, in Fig 5, the X-axis label is written in Portuguese, not English.

Answer: The grammar and vocabulary have been revised.

Round 2

Reviewer 1 Report

Authors did not take into account all my comments.

In particular, I never saw a clear explanation why they did not calculate catalytic constants such as Km, Vmax and Kcat. These parameters are among the most important characteristics of immobilized enzymes.

Secondly, authors added in article a table  containing the results of comparison of the optimized enzyme loading and activity recovery, as I recommended. But they did not discuss in the text the results from this table. For example, in Ref. 42 the retention of activity was 150%. Then why is the immobilized lipase described in this article better? It is not clear why the authors included in the table the results obtained for catalases (Ref. 40). Moreover, it is better to include these arguments in the text of the article, and not in supplementary materials.

Authors should clearly state in the article what advantages their immobilized lipase has over immobilized lipases obtained by other scientists, including lipases from other producers and lipases immobilized on other carriers. 

Author Response

DETAILED ANSWER TO THE REVIEWER’S COMMENTS

REVIEWER 1

Authors did not take into account all my comments.

In particular, I never saw a clear explanation why they did not calculate catalytic constants such as Km, Vmax and Kcat. These parameters are among the most important characteristics of immobilized enzymes.

Secondly, authors added in article a table  containing the results of comparison of the optimized enzyme loading and activity recovery, as I recommended. But they did not discuss in the text the results from this table. For example, in Ref. 42 the retention of activity was 150%. Then why is the immobilized lipase described in this article better? It is not clear why the authors included in the table the results obtained for catalases (Ref. 40). Moreover, it is better to include these arguments in the text of the article, and not in supplementary materials.

Authors should clearly state in the article what advantages their immobilized lipase has over immobilized lipases obtained by other scientists, including lipases from other producers and lipases immobilized on other carriers.

Answer: We fully agree with the reviewer that the calculation of catalytic constants such as Km, Vmax and Kcat are important parameters for immobilized enzymes. However, due to the holiday season and extremely short time for reviewing the manuscript, we are unable to perform these parameters. However, we appreciate the Reviewer's comment and, in our next study on the application of immobilized lipase, the calculation of catalytic constants will be properly planned and performed. We would like to highlight that the main focus of this study was to evaluate the new support, as we can also observe in other works in the literature in which the focus was on the support not performing the calculations of the catalytic constants ( https://doi.org/10.1016/j.carbon.2010.09.037 , https://doi.org/10.1021/la202794t,  https://doi.org/10.3390/ma14112874 ).

Secondly, the table was added in the text of the article, and further discussion was added in the text as the explanation for the 150% activity retention of Ref 42. In addition to further work on immobilization of lipases on other supports.

Round 3

Reviewer 1 Report

The authors took into account most of my comments.

Only one of them was not answered: "The performed study should be a connected with determined enzyme catalytic constant such as Km, Vmax and Kcat".

Taking into account the specifics of the journal, I think that the kinetic characteristics can be left for further research, and the article can be accepted for publication in the presented form.